# Ectromelia Virus Affects the Formation and Spatial Organization of Adhesive Structures in Murine Dendritic Cells In Vitro

**DOI:** 10.3390/ijms25010558

**Published:** 2023-12-31

**Authors:** Zuzanna Biernacka, Karolina Gregorczyk-Zboroch, Iwona Lasocka, Agnieszka Ostrowska, Justyna Struzik, Małgorzata Gieryńska, Felix N. Toka, Lidia Szulc-Dąbrowska

**Affiliations:** 1Department of Preclinical Sciences, Institute of Veterinary Medicine, Warsaw University of Life Sciences, 02-786 Warsaw, Poland; zuzanna_biernacka@sggw.edu.pl (Z.B.); karolina_gregorczyk-zboroch@sggw.edu.pl (K.G.-Z.); justyna_struzik@sggw.edu.pl (J.S.); malgorzata_gierynska@sggw.edu.pl (M.G.); ftoka@rossvet.edu.kn (F.N.T.); 2Department of Biology of Animal Environment, Institute of Animal Science, Warsaw University of Life Sciences, 02-786 Warsaw, Poland; iwona_lasocka@sggw.edu.pl; 3Department of Nanobiotechnology, Institute of Biology, Warsaw University of Life Sciences, 02-786 Warsaw, Poland; agnieszka_ostrowska@sggw.edu.pl; 4Department of Biomedical Sciences, Ross University School of Veterinary Medicine, Basseterre P.O. Box 334, Saint Kitts and Nevis

**Keywords:** Ectromelia virus, dendritic cells, podosomes, focal adhesions, migration

## Abstract

Ectromelia virus (ECTV) is a causative agent of mousepox. It provides a suitable model for studying the immunobiology of orthopoxviruses, including their interaction with the host cell cytoskeleton. As professional antigen-presenting cells, dendritic cells (DCs) control the pericellular environment, capture antigens, and present them to T lymphocytes after migration to secondary lymphoid organs. Migration of immature DCs is possible due to the presence of specialized adhesion structures, such as podosomes or focal adhesions (FAs). Since assembly and disassembly of adhesive structures are highly associated with DCs’ immunoregulatory and migratory functions, we evaluated how ECTV infection targets podosomes and FAs’ organization and formation in natural-host bone marrow-derived DCs (BMDC). We found that ECTV induces a rapid dissolution of podosomes at the early stages of infection, accompanied by the development of larger and wider FAs than in uninfected control cells. At later stages of infection, FAs were predominantly observed in long cellular extensions, formed extensively by infected cells. Dissolution of podosomes in ECTV-infected BMDCs was not associated with maturation and increased 2D cell migration in a wound healing assay; however, accelerated transwell migration of ECTV-infected cells towards supernatants derived from LPS-conditioned BMDCs was observed. We suggest that ECTV-induced changes in the spatial organization of adhesive structures in DCs may alter the adhesiveness/migration of DCs during some conditions, e.g., inflammation.

## 1. Introduction

Ectromelia virus (ECTV) belongs to the genus *Orthopoxvirus* of the family *Poxviridae* [1,2]. Since the 1940s, it has been considered by scientists worldwide as an excellent model to study the pathogenesis and immunology of orthopoxvirus infection [3], including variola virus (VARV), which has decimated humanity in the past [4,5]. Although VARV, which has now been eradicated through vaccination programs [4,6,7], is considered a problem of the past, it can nevertheless pose a major threat as a potential biological weapon [8,9]. Additionally, an increasing number of zoonotic poxvirus infections, including human monkeypox virus (MPXV) infection outbreaks in endemic (Central and West Africa) and non-endemic (North America, Europe) regions [10], vaccinia virus (VACV) infections in humans in India and South America [11,12], and cowpox virus (CPXV) infections in pets and humans, especially in Europe [13], cause a serious threat [10] to human health worldwide.

MPXV, VACV, and CPXV have a large spectrum of hosts, whereas VARV and ECTV are restricted to only one host species (human and mouse, respectively). Such differences in the host range restriction are caused by the presence/absence of different host range genes, which ultimately determine the different pathogenesis, tropism, and immune response regulation between polyhostal and monohostal viruses [14]. ECTV is highly restricted to its natural host—the mouse; therefore, it is a suitable model for studying both the immune escape mechanisms of the virus and the antiviral immune response of the host in the virus-host coevolutionary system. As a consequence of host restriction, ECTV is the only known orthopoxvirus that is able to productively replicate in dendritic cells (DCs)—highly specialized cells of the immune system [15,16]. 

DCs are antigen-presenting cells (APCs) whose role is to capture and then present antigens to naïve T lymphocytes in secondary lymphoid organs [17,18,19]. The interaction of DCs with the antigen leads to their activation and maturation, which is accompanied by many morphological, phenotypic, and physiological changes [20]. Mature DCs express increased levels of major histocompatibility complex (MHC) and costimulatory molecules [21,22,23,24], as well as change their adhesion phenotype and migratory properties [25].

Cell migration, enabled by cytoskeleton protein rearrangement, is the most essential mechanism responsible for the rapid detection of a threat and the induction of an immune response [26]. Many complex adhesive structures, such as podosomes or focal adhesions (FAs), are involved in the migration of immature DCs, which are constantly tasked with controlling the pericellular environment [27,28,29]. Podosomes are actin-based mechanosensitive adhesion structures that form the podosome core, together with gelsolin, cortactin, the Arp2/3 complex, and the Wiskott-Aldrich syndrome protein (WASP). The podosome core is surrounded by a ring of integrins and adaptor proteins (including vinculin, talin, and paxillin) that connect the cortical actin cytoskeleton to the cell membrane and integrins [28,30,31]. Podosomes form junctions between the cell and the extracellular matrix (ECM) and enable slow mesenchymal migration of immature DCs due to the release of proteolytic enzymes that locally degrade the ECM [32,33]. The dissolution of podosomes occurs during DC maturation and is associated with a rapid ameboid mode of migration [34].

FAs are complex macromolecular assemblies associated with the plasma membrane that enable cell adhesion to the substrate and migration [35,36]. The formation and dynamics of FA maturation are therefore central to many biological processes [37]. FAs link the actin cytoskeleton to integrins, which are linkers to the ECM, and modulate their binding strength, thereby controlling actin organization. Like podosomes, FAs are composed of actin, adaptor proteins (for example, vinculin and talin), and signaling proteins, such as FAK kinases, phosphatases, phospholipases, or regulators of the small guanosine triphosphatase (GTPase) [35,38,39]. Despite many similarities, podosomes and FAs differ in their dynamics and tension. Podosomes are considered to be more dynamic, degradable, and less stable structures compared with FAs, but importantly, any change within these structures is important for the migratory function of the cell [40].

Because assembly and disassembly of adhesive structures are highly associated with DCs’ immunoregulatory properties and migratory function, we evaluated how ECTV productive infection targets podosome and FA organization and formation in natural-host bone marrow-derived DCs (BMDCs). We found that ECTV induces a rapid dissolution of podosomes at the early stages of infection (4 h post-infection (hpi)), accompanied by the development of larger and wider FAs than in control uninfected cells. Dissolution of podosomes in infected cells was not associated with the maturation of BMDCs, since at 24 hpi, BMDCs exhibited decreased surface expression of MHC II, CD80, and CD86 molecules. FAs were predominantly observed within different types of long cellular projections formed extensively during the later stages of infection (18–24 hpi). Interestingly, loss of podosomes does not influence the migratory properties of infected BMDCs in a wound healing assay but accelerates their transwell migration towards LPS-conditioned medium. We suggest that ECTV modulates adhesive structure organization in DCs to alter their adhesiveness/migration in some, e.g., inflammatory conditions.

## 2. Results

### 2.1. Immature BMDCs Exhibit the Presence of Different Types of Podosomes

Depending on the cell type and environmental conditions, podosomes can take various forms, from simple single structures to more complex podosome superstructures, such as clusters, belts, rings, and rosettes [41,42]. Single podosomes are individual structures distributed at visible intervals on the bottom of the cell and show no apparent superstructure [41]. Podosome clusters are classified as groups of more than ten individual podosomes forming together in a restricted area of the cell [43]. In osteoclasts, podosome clusters evolve to transient ring patterns and eventually to more massive circular structures, such as podosome belts or sealing zones [44]. Podosome rosettes are different-shaped superstructures (circular, concave, or ellipsoidal) with a diameter of 5–20 µm, which contain highly concentrated adhesion proteins and are very potent in matrix degradation [42].

In immature BMDCs cultured on microscopic slides, podosomes had a typical structure, i.e., a podosome core composed of F-actin and a ring surrounding the core composed of the adaptor protein vinculin (Figure 1). Based on the morphological organization, podosomes in BMDCs were classified into the three most common types: single, clusters, and rosettes. All three forms differed in the degree of complexity of their organization, from the simplest single forms to the most complex superstructures—rosettes [41,42]. Individual podosomes with a loosely connected structure and distributed far from each other or separated by empty space (containing no adhesive proteins) were classified as single (Figure 1). The fluorescence intensity in cells with single podosomes oscillated between fluorescence decay (in unstained areas, thus showing no podosomes) and high fluorescence levels at the site of the podosome (vinculin—the ring surrounding the actin—core). Single podosomes tended to occur evenly throughout the bottom surface of the cell or in particular parts of the cell bottom.

Podosome clusters were classified as having a dense, well-organized structure of podosomes, with no empty space between individual podosomes (Figure 1). Such a connected network of individual podosomes formed one large group or several smaller groups on the bottom surface of the cell—often at the leading edge of the cell, which may be directly related to cell migration. Moreover, fluorescence in the podosome clusters remained consistently high throughout, confirming the close proximity of the podosomes and their high density. Podosome clusters were observed more frequently in immature BMDCs cultured in a dense 2D environment.

Podosome rosettes were classified as superstructure forms characterized by large ring-shaped bands composed of vinculin surrounding, on both sides, a central band composed of dense actin dots. Rosettes had a high level of fluorescence in the ring (from actin and vinculin) and a lack of fluorescence inside the ring (Figure 1).

### 2.2. ECTV Infection Induces Podosome Dissolution in BMDCs

In uninfected control BMDCs, the presence of podosomes was observed in each measurement window (Figure 2). Control cells assumed a regular shape, and podosomes were usually present in the peripheral part of the cells at their leading edge (Figure 2A). The podosomes in the control cells were characterized by a well-organized ensemble structure, especially in the late hours of culture (Figure 2B). At each hour, podosome clusters were the most numerous group of podosomes (around 50%). At the 4th hour of culture, more than 40% of the cells had single podosomes in a loosely arranged group, but with time of incubation, more compact structures were observed (Figure 2A). It suggests that substrate adhesion is an important determinant of podosome formation—at 18 hpi, more than 30% of the cells had rosette podosomes, while at 24 hpi, more than 40% (Figure 2B). Less than 5% of control cells had no podosomes and showed only the presence of FAs. The above trend was observed at each of the analyzed time points (4, 8, 12, 18, and 24 h).

ECTV productively infects BMDCs with the extracellular release of progeny viral particles at later stages of infection [16]. Because ECTV is a large dsDNA virus that replicates within the cytoplasm, the course of infection can be easily monitored using a fluorescence microscope after staining the viral dsDNA using Hoechst 33342 and/or viral proteins with specific FITC-labeled antibodies (Figure 2A) [16]. The first visible sign of cellular infection at 4 hpi is the presence of small, regular intracytoplasmic viral factories (virus replication sites), located near the cell nucleus (Figure 2A). At 8 hpi, viral factories increase in size but still remain regular. At 12 hpi, single virus particles can already be observed next to the viral factories, evenly distributed in the cytoplasm. In the late stages of infection (18–24 hpi), large, sometimes irregular “bloated” virus factories and numerous progeny virus particles are found in the cytoplasm, also within the emerging long cellular projections (Figure 2A) [16].

In ECTV-infected cells, podosome dissolution was observed as early as 4 hpi (Figure 2A) and was occurring in cells at the time of viral factory formation (Figure 2A). At 4 hpi, more than 60% of ECTV-infected cells had no podosomes, and by 8 hpi, this trend had increased to about 90% (at 8 hpi, a rapid disappearance of podosomes was observed). After 8 hpi, the appearance of long FAs in ECTV-infected cells was observed (Figure 2A). At 8, 12, 18, and 24 hpi, the cells begin to assume a characteristic shape with long, actin-based cellular protrusions (Figure 2A). In addition, at the late stages of infection (after 12 hpi), progeny viral particles were observed in the vicinity of the viral factory, colocalizing at the actin filaments in the long cellular protrusions (Figure 2A) and forming progeny viral factories. In the late stage of ECTV infection (18 hpi), no podosomes were observed in more than 95% of cells, and in 24 hpi—in almost 100% (Figure 2B). In a small percentage of ECTV-infected cells in which podosomes were present, no rosette podosomes were observed (Figure 2B). 

In LPS-stimulated cells (which represent a control for cell maturation), a gradual disappearance/dissolution of all podosome types and the appearance of numerous FAs were observed as early as 4 h after stimulation (Figure 2A). This trend persisted until 24 h after stimulation. In addition, after LPS stimulation, the cells changed to a more regular shape and were larger relative to control cells, which was associated with a rearrangement of actin in the cell. At each hour of the time points analyzed (4, 8, 12, 18, and 24 h), single ungrouped podosomes or podosome clusters were observed in approximately 40% of the cells, but no podosome rosette formation was observed (Figure 2B). 

In the case of ECTV infection of cells with simultaneous LPS stimulation (ECTV + LPS), a similar trend to that observed for ECTV-infected cells was observed, i.e., the disappearance of podosomes correlated with the formation of a viral factory (Figure 2A). At the same time, in the later hours of cell culture (12, 18, and 24 hpi), there was a change in cell shape to an irregular shape with many actin protrusions, at the end of which distinct FAs were observed (Figure 2A). Moreover, in cells treated with ECTV + LPS, the disappearance of podosomes at 4 hpi was more pronounced than in cells with ECTV infection alone (Figure 2B); however, cells at 8 hpi with ECTV infection compared with ECTV + LPS were characterized by a greater abundance of cells without podosomes. Thus, BMDCs, in which the viral factory has been well established, showed virtually complete disappearance of podosomes and the appearance of FAs, especially in the later stages of ECTV infection, regardless of LPS treatment (Figure 2A,B). Taken together, these data indicate that ECTV induces podosome dissolution in BMDCs at the early stages of infection.

### 2.3. Podosome Dissolution in ECTV-Infected BMDCs Is Not Associated with Cell Maturation

Podosomes are a characteristic feature of immature DCs; therefore, in the next step, we assessed the relationship between the dissolution of podosomes and cell maturation in ECTV-infected BMDCs. For this purpose, the level of MHC class II molecules, as well as costimulatory molecules CD80 and CD86, was analyzed in control cells and cells treated with ECTV, LPS, or ECTV + LPS after 24 h exposure. 

In ECTV-infected cells, MFI for MHC II, CD80, and CD86 decreased significantly (*p* < 0.05) compared with uninfected control BMDCs (Figure 3). LPS treatment of uninfected cells for 24 h resulted in cell maturation since MFI for all analyzed markers increased more than 2-fold on the BMDC surface. Meanwhile, BMDCs infected with ECTV and treated with LPS were unable to increase the expression of MHC II, CD80, and CD86 molecules to the levels observed in cells treated only with LPS, and the inhibitory effect of the virus on MHC II, CD80, and CD86 levels was more effective compared with cells treated with ECTV alone. In conclusion, BMDCs after ECTV infection are not able to fully mature and retain an immature phenotype, and therefore podosome dissolution is not associated with cell maturation. 

### 2.4. FAs Grow in Size in BMDCs Infected with ECTV

In the next step, we evaluated the dynamics of formation and morphometric parameters of FAs in BMDCs during the replication cycle of ECTV. FAs, like podosomes, are actin-based structures that connect the ECM to the cytoskeleton via membrane integrin receptors and are structures that recruit proteins, such as vinculin and paxillin, for their formation (Figure 4A). FAs, as structures from which many biological responses to external forces originate, were observed in BMDCs under all experimental conditions tested in each measurement window; however, there were differences in their number, length, and width between particular experimental groups (Figure 4B). In uninfected control BMDCs, the number of FAs/cell was stable during 24 h of incubation and reached the mean values of 29 and 28 at 4 and 24 hpi, respectively. ECTV infection resulted in a statistically significant (*p* < 0.05) increase in the number of FAs/cell only during the later stages of infection (24 hpi) and reached the mean value of 32 (Figure 4B). The change in the number of FAs/cell was significantly discernible when cells were treated with LPS at both 4 and 24 hpi (*p* < 0.001) and reached the mean values of 69 and 64, respectively (Figure 4B). Simultaneously, ECTV-infection of LPS-treated cells resulted in a decrease in the number of FAs/cell at both time measurements (*p* < 0.05), compared with uninfected LPS-treated cells, and the mean values of 43 and 54 at 4 and 24 hpi, respectively (Figure 4B). Statistical analysis revealed that there was a significant (*p* < 0.001) increase in the number of FAs/cell in the ECTV-infected LPS-treated BMDC group at 24 hpi compared with 4 hpi (Figure 4B).

The virus had the most significant effect on the length and width of FAs in BMDCs during the whole virus replication cycle (Figure 4B). At 4 and 24 hpi with ECTV, the length of FAs significantly increased compared with control cells and reached the mean values of 1.48 µm vs. 1.25 µm at 4 hpi and 1.96 µm vs. 1.14 µm at 24 hpi. It can also be seen that the length of FAs in ECTV-infected cells was higher at 24 hpi compared with their counterparts at 4 hpi, and this increase was statistically significant (*p* < 0.001). An identical analogy concerned the width of FAs. Meanwhile, LPS treatment of uninfected cells does not influence the length of FAs, however slightly, but significantly (*p* < 0.001) increased the width of FAs at 4 (0.27 µm vs. 0.24 µm) and 24 (0.28 µm vs. 0.25 µm) hpi, compared with control unstimulated cells. Similarly to ECTV-treated LPS-unstimulated cells, ECTV-infection of LPS-treated cells resulted in a significant increase in the length of FAs at 4 (1.42 µm vs. 1.25 µm) and 24 (1.93 µm vs. 1.18 µm) hpi, compared with LPS-treated uninfected cells. The width of FAs was significantly elevated in cells treated with ECTV + LPS compared with cells treated with LPS alone only at 24 hpi (Figure 4B). In all experimental conditions, the length of FAs increased significantly with the time of analysis, and at 24 hpi, the mean values were higher compared with 4 hpi. The width size increase between 24 hpi and 4 hpi was only observed in both ECTV-infected groups (Figure 4B). Meanwhile, correlation analysis between the length and width of FAs did not show any significant dependence in each of the experimental groups (Figure 4C). Therefore, the length of FAs does not correspond to their width. In conclusion, the virus significantly affects the number of FAs in infected BMDCs during the later stages of infection but significantly elevates the length and width of FAs during the whole replication cycle, with a higher tendency at 24 hpi.

### 2.5. FAs Are Formed Preferentially in Long Cellular Extensions of ECTV-Infected BMDCs

Next, we explored the spatial distribution of FAs in ECTV-infected BMDCs. In control and LPS-stimulated uninfected cells, FAs were observed on the entire cell periphery of the ventral surface; however, in the latter cells, FAs were more dense than those formed by control cells. Such FA distribution was observed in cells at 4 and 24 hpi (Figure 4A). Meanwhile, ECTV-infected cells exhibited the presence of FAs almost exclusively within the cellular extensions formed extensively at later stages of infection (since 12 hpi) (Figure 4A). As early as 4 hpi, when regular viral factories started to appear within the cytoplasm of infected cells and podosomes started to dissolute, the cells began to change their morphology, and FAs began to accumulate in regions from which long actin projections would form at a later stage of the virus replication cycle (Figure 5). At 8 and 12 hpi, cellular extensions began to elongate, and FAs were located mainly in the distal parts of these extensions. In the late stages of infection (18–24 hpi), the cells had peculiar, branched shapes due to the presence of elongated actin extensions, in which FAs were larger and were located mainly in their distal parts, as well as in the places where the extensions branched (Figure 5). 

FAs are present in different types of long cellular extensions, formed extensively by ECTV-infected BMDCs at the later stages of infection (Figure 6). Among different morphological forms of the protrusions, four characteristic types could be observed, i.e., thin (Figure 6A), thickened at the ends (Figure 6B), wide (Figure 6C), or branched (Figure 6D). Long, thin projections were not extended at the ends, and their length often exceeded the length of the cell body (Figure 6A). The projections with thickened ends were usually thin (or slightly thickened) but flared at the end. The flared end often contained mitochondria (Figure 6B). Long, wide projections were even 10-times wider than thin cellular projections and contained relatively evenly distributed mitochondria (Figure 6C). Branched projections showed a complex branching pattern. These projections were the longest, and their length often exceeded the cell body even two times (Figure 6D). In each of these types, the presence of viral particles, mitochondria, tubulin fibers, and often increased acetylation of α-tubulin could be observed. The progeny viral particles co-localized with actin filaments in the protrusions, which formed a kind of ‘cytoplasmic package’ containing and having the capacity to release numerous progeny virions from the cell. The viral progeny particles also co-localized with microtubules, which the virus used to transport to the periphery of the cell. The long cellular protrusions also contained numerous, small, and distinct mitochondria, for which the scaffolding was a network of microtubule filaments (Figure 6). Together, these data indicate that long cellular extensions play a role in the active spread of progeny viral particles. Meanwhile, thin cellular extensions usually contained one long FA at the distal tip (Figure 6A), while thickened end projections had several (usually two or three) FAs at the distal end (Figure 6B). The wide cellular extensions contained many smaller FAs distributed throughout the entire surface and branched projections that had many longer FAs at the ends or in the branching regions (Figure 6C). However, detailed analysis of differences in FA patterns in various types of long cell projections and their potential role in virus spreading and adhesion-mediated signaling pathways requires further experiments.

ECTV particles localizing to the long protrusions also induced the formation of short protrusions—actin tails—which pushed them out of the cell (Figure 7A). The formation of FAs in actin tails was not observed. Moreover, FA occurrence in another type of short actin projection—dendrites—was relatively rare (Figure 7B).

### 2.6. ECTV Infection Does Not Affect BMDC Migration in the Wound Healing Assay

Because podosomes and FAs contribute to cell migration, in the next step, we assessed if changes in the organization of adhesive structures may affect the cell migration rate of ECTV-infected BMDCs. To examine this purpose, the wound-healing assay and the Transwell-migration assay were applied. In the wound healing assay, images were recorded on the same fields of confluent cells immediately after the scratch (time 0) and after 24 h of incubation. Since single cells were already observed in the wound area at time 0, we assessed wound closure by determining the number of cells migrating into the wound after 24 h of incubation. After 24 h, there were no significant differences in migration rates between uninfected and ECTV-infected cells, with in both cases less than 100 cells migrated into the wound area (Figure 8A,B). Meanwhile, LPS stimulation caused a significant increase in the rate of cell migration compared with control cells not treated with LPS, and an average of over 150 cells entered the wound area (Figure 8A,B). Cells co-treated with ECTV + LPS acquired slightly weaker migratory ability compared with uninfected cells stimulated with LPS, where approximately 140 cells entered the wound; however, this slight difference was not statistically significant (Figure 8A,B). In the wound area, even control uninfected cells showed virtually no podosomes, which demonstrated their migratory potential (Figure 8C). These data demonstrate that, despite inducing podosome dissolution and promoting the formation of longer and broader FAs, ECTV does not significantly affect BMDC migration under 2D conditions in vitro.

### 2.7. ECTV Infection Accelerates the Directional Migration of BMDCs toward the LPS-Conditioned BMDC Supernatants

The transwell migration test was used to examine the directional migration of ECTV-infected BMDCs towards the chemoattractant, which was the post-culture supernatant from BMDCs stimulated for 24 h with LPS. The migrating cells were imaged on the bottom of the transwell migration membrane and quantified by counting the cells that had migrated through a porous membrane into the bottom well using an inverted microscope. Our results showed that this time ECTV-infected BMDCs showed a significantly (*p* < 0.05) higher migratory rate towards the chemoattractant compared with the uninfected control cells (Figure 9A,B). Meanwhile, cells treated with ECTV + LPS had comparable migratory rates to LPS-treated uninfected cells (Figure 9A,B). These data suggest that ECTV may reduce adhesion and/or increase migration of BMDCs during some inflammatory conditions.

## 3. Discussion

ECTV is an orthopoxvirus that constitutes a convenient model for studying changes occurring in DC physiology and function because this is a large virus that evolved with its natural host (the mouse) and has developed several mechanisms that allow it to completely subordinate immune cells to its replicative needs [45,46,47]. Therefore, a better understanding of the interactions between ECTV and the formation of adhesive structures, especially in specialized cells of the immune system, will significantly facilitate understanding of the mechanisms regulating the basic innate functions of DCs, such as migration, which plays a key role in the pathogenesis of several infectious and non-infectious diseases and, on the other hand, is indispensable in maintaining immune surveillance and tissue homeostasis.

Migration is an elementary mechanism that ensures that DCs function properly biologically as APCs. The migration and adhesion phenotype of DCs changes not only during immune surveillance, tissue homeostasis, or pathological conditions but is also altered during DC differentiation and maturation. The migration and scanning of the DCs’ environment are facilitated by specific adhesive structures—podosomes and FAs. Podosomes are mechanosensory elements with proteolytic properties, composed of multiple proteins, including vinculin and paxillin, that link integrins to actin filaments, enabling DCs to penetrate the ECM and ‘scan’ the environment for antigens. Podosomes localize at strategic locations in the cell (at the contact surface), usually in the anterior part of the cell, the so-called lamellipodium [48]. Our study revealed that uninfected BMDCs form three basic types of podosomes on a glass slide, i.e., single, clusters, or rosettes, the two latter of which remain in close association with each other. The number of rosette-type podosomes increased with the time of cell culture, and the largest number of them could be found after 24 h of culture on the glass slide. Therefore, the culture time promotes the formation of rosette podosomes, which are the most complex podosome superstructures with the highest capacity to promote matrix degradation [49]. It has been shown that podosome rosettes are generated through one of two mechanisms: de novo assembly, which involves several protein tyrosine kinases, such as Src and FAK [50,51], or fission of pre-existing podosome rosettes in Src-transformed fibroblasts [49]. Interestingly, fission is more efficient than de novo assembly in the generation of new podosome rosettes, and podosome rosettes undergoing fission display higher motility and matrix-degrading capacity than non-fission podosome rosettes [49]. 

In the present study, we showed that ECTV infection leads to rapid podosome dissolution in the early stages of ECTV infection. The virus induced podosome dissolution in more than 60% of the observed cells, as early as the onset of the viral factory at 4 hpi. If podosomes remained in the cells, they took the form of single structures, practically without the presence of podosome superstructures like clusters or rosettes. From 8 hpi on, virtually no podosomes were observed in infected cells, and this tendency persisted until the late stages of the replication cycle, i.e., 24 hpi. Therefore, it is not excluded that ECTV may negatively regulate podosome-mediated functions of DCs, such as migration, adhesion, antigen uptake, and ECM degradation. Similarly to our results, the measles virus (MV) has also been shown to induce podosome atrophy in DCs, which was associated with the acquisition of a fast ameboid mode of migration by these cells in a 3D environment [52]. Ameboid migration mode was probably promoted by the sphingosine kinases (SphK)/sphingosine 1-phosphate (S1P) system, characterized not only by podosome dissolution but also by the loss of filopodia [52]. Studies on DCs have shown that invasion by the pathogenic protozoan *Toxoplasma gondii* also affects actin rearrangement and podosome dissolution. At the same time, an increased migratory potential of these cells was observed [53]. Moreover, a report from 2017 [54] indicated that *T. gondii* changes the migration mode of DCs to a rapid amoeboid mode, accompanied by podosome dissolution and up-regulated secretion of tissue inhibitor of metalloproteinases-1 (TIMP-1), which impaired ECM proteolysis. 

Because DC maturation has been shown to induce podosome dissolution [27,55], we thus tested if ECTV infection, which induced podosome atrophy, also induced DC maturation. LPS-treated DCs were used as a control of fully mature cells, as confirmed by increased expression of MHC II, CD80, and CD86 molecules, and indeed, such cells exhibited podosome loss, especially after 24 h of incubation with LPS. Meanwhile, ECTV-infected DCs were unable to reach full maturity, even after LPS treatment. These results indicate that podosome disassembly induced by ECTV does not require DC maturation. It should be mentioned that the relationship between the maturation status and disappearance of podosomes in DCs is not so obvious and largely depends on the signaling pathways triggered during DC stimulation. For example, LPS-triggered TLR4-mediated maturation of DCs induces the loss of podosomes, whereas Pam_3_CSK_4_-triggered TLR2-mediated maturation maintains podosome formation and stability [56]. Additionally, Gram-negative bacteria, such as *Neisseria meningitidis* or *Salmonella enteritidis*, induce podosome disassembly and transform adhesive DCs into highly migratory cells in an LPS- and TLR4-dependent manner. Gram-positive bacteria, such as *Staphylococcus aureus* or *Streptococcus pneumoniae*, failed to induce podosome dissolution and did not alter the migratory properties of DC [55]. These results suggest that different infectious agents may activate distinct TLRs, which may result in DC maturation but with different effects on podosome formation.

In addition to podosomes, FAs are also involved in migration due to their ability to respond to the mechanical properties of the tissue microenvironment and incoming signals from the environment to the cells. The resultant events manipulate the composition of FA-building proteins so that specific biochemical signals are transmitted that mediate cellular behavior [35]. Despite many similarities, podosomes and FAs differ in their dynamics and tension. Podosomes are considered to have more dynamic, degradable, and less stable structures compared with FAs, but importantly, any change within these structures is important for the migratory function of the cell [40]. 

Our study revealed that ECTV-infected DCs contained longer and thicker FAs compared with control, uninfected cells. The number of FAs was unaffected in infected cells. FAs were present mainly in long cellular protrusions formed extensively by ECTV-infected cells. It is not known, however, how large FAs may affect the movement of ECTV-infected DCs. It has previously been postulated that there is an inverse relationship between the FA size and the cell migration rate, since observations in different cell types revealed that fast-moving cells (e.g., *Dictyostelium discoideum*, human neutrophils, fish keratocytes) show vanishingly small FAs at their basal surface [57,58,59], whereas slow-moving cells (e.g., fibroblasts, endothelial cells) display large FAs [60]. However, more recent data on the functional relationship between FA morphometric parameters and cell migration revealed a completely opposite relationship, as it was shown that focal adhesions were larger and more elongated in fast-moving mouse embryonic fibroblasts (MEFs) [61]. More specifically, focal adhesion size (but not their number, surface density, shape, or molecular composition) affects cell speed biphasically. During the first phase, MEF velocity increases steadily with focal adhesion size until an optimum is reached (a threshold of ~0.7, corresponding to ~2.6 μm^2^ in the normalized focal adhesion size). During the second phase, cell migration decreases after exceeding the maximum threshold (>2.6 μm^2^) as a result of the increasing size of the FAs [61,62]. Therefore, it is suggested that the size of FAs can be a highly predictive factor of cell migration speed [61,62]. Additionally, increased size of FAs has been shown to elevate adhesion forces, but only during the initial stages of myosin II-mediated FA maturation. After this period, the FA size remains constant, and mature adhesions can withstand even sixfold increases in tension without changes in size [63]. This indicates that there is no correlation between traction force and FA size in mature, elongated FAs. Rather, the proximity to the cell edge influences the tension that is sustained at mature FAs, and higher tension is transmitted within peripheral adhesions, while lower tension is observed in adhesions near the cell center [63]. Using a chemomechanical model, it has also been found that larger FAs with higher levels of contractile force are induced by a stiffer substrate, and with increasing substrate rigidity, more adhesions are predicted to form due to the decreasing nucleation size [64]. 

In our study, ECTV-infected cells, despite the appearance of longer and thicker FAs, did not show changes in the rate of 2D migration. Moreover, in infected cells, podosomes were dissolved, which hypothetically could also enhance the migration of these cells. On the other hand, LPS treatment of uninfected cells induced podosome dissolution, increased the formation of high numbers of FAs without affecting their size, and increased cell motility in a 2D environment. This result is consistent with the study by West et al. (2008) [65], which showed that loss of podosomes, induced by LPS binding by TLR4, resulted in pronounced FA formation and was dependent on ADAM17 (a disintegrin and metalloprotease 17). It is worth mentioning that LPS-treated DCs showed faster migration rates after prolonged exposure to LPS; however, short exposure to LPS (up to 2 h) induced a transient inhibition of DC migration, probably needed for enhanced actin-dependent uptake of local antigens [65]. In ECTV-infected cells, in addition to the switch from podosomes to FAs, the appearance of long actin projections was also observed. In long cellular projections, progeny viral factories and progeny viral particles co-localized with actin filaments. This is because the long projections formed specific cytoplasmic corridors that enabled the mass release of progeny virions from the cells [66]. In addition, mitochondria, which can provide the energy necessary for viral replication and transport, are co-localized with tubulin filaments near viral particles within the projections. Moreover, long cellular protrusions contained numerous FAs at their ends. Taken together, the appearance of distinctive cellular protrusions and the increase in length and maturity of focal adhesion at the protrusion tips of ECTV-infected BMDCs could substantially increase the contact area and adhesion strength of cells to the substrate, thus reducing the ability of cells to migrate, especially in the later stages of infection. On the other hand, during cell handling, we observed that ECTV-infected cells always detached from the glass or plastic surface more easily than LPS-treated or control cells. 

In contrast to 2D migration, ECTV-infected cells showed an increased migration rate in a transwell assay towards the medium containing chemotactic compounds derived from LPS-stimulated BMDCs. It is an interesting result, as our previous studies [16] as well as those of other authors [67] have shown that ECTV reduces the expression of receptors for inflammatory cytokines/chemokines on DCs and impairs DC migration [68]. However, it is not excluded that the tempo of cell migration depends on the stage of the virus replication cycle in BMDCs. In the early stages of infection, cells could respond to chemoattractants and move faster, but later in infection, when the cells developed long cellular projections, their movement could be significantly slowed down. VACV has been demonstrated by a wound healing assay to induce two forms of cell movement: (i) fast independent migration of infected cells into the wound (induced by early viral proteins) and (ii) slow migration of cells after the development of long-branched projections (induced by late viral proteins) [69]. Orthopoxviruses encode numerous genes involved in actin cytoskeleton rearrangement, interactions of integrins with the ECM, and cell adhesion [70,71,72,73,74,75], which subsequently may affect cell migration. Therefore, future studies are needed to elucidate the effect of adhesive structure rearrangement on the migration of DCs in the context of ECTV infection.

To summarize, ECTV in DC causes changes in the organization of adhesion structures, affecting podosomes and focal adhesions already in the initial stages of infection. This virus constitutes a valuable research model for studying the molecular mechanisms involved in the rearrangement of the cytoskeleton, the formation of adhesive structures, and the migration of these specialized immune system cells. Since the transformation of adhesive structures in DCs plays an important role in the rate of migration of these cells through the extracellular matrix to the lymph nodes to present antigens to T lymphocytes, elucidating their role in orthopoxvirus infection will contribute to a better understanding of the mechanisms of pathogenesis of systemic diseases.

## 4. Materials and Methods

### 4.1. Virus

A highly virulent and infectious strain of Moscow ECTV (ECTV-MOS, VR-1374; American Type Culture Collection (ATCC), Manassas, VA, USA) was used in the experiments. The virus was propagated and titrated on African green monkey kidney epithelial cells (Vero) (ATCC, CCL-81), as described previously [15]. 

### 4.2. Animals

Male C57BL/6 (H-2b) mice were purchased from the animal facility at the Maria Skłodowska-Curie Institute—Oncology Centre in Warsaw. After importation, the animals underwent a period of acclimatization for 7 days in the animal facility at the Institute of Veterinary Medicine under controlled temperature and humidity and access to food and water ad libitum. Animals from 8 to 12 weeks of age were used in the experiments for tissue collection; thus, the approval of the Local Ethical Committee was not required. All experiments were conducted according to institutional guidelines for the care and use of laboratory animals.

### 4.3. Obtaining a BMDC Culture

Primary culture of BMDCs was obtained using mouse recombinant granulocyte-macrophage colony stimulating factor (rmGM-CSF; Sigma-Aldrich, St. Louis, MO, USA), as described previously [45] with minor modifications. Bone marrow isolated from the tibia and femur was suspended in RPMI 1640 medium (Gibco, Grand Island, NY, USA) supplemented with 1% antibiotic (100 U/mL penicillin and 100 µg/mL streptomycin, Symbios, Gdansk, Poland). Erythrocytes were lysed with NH_4_Cl-Tris buffer. Cells were resuspended in R-10 growth medium [RPMI-1640 medium enriched with 10% heat-inactivated fetal bovine serum (FBS, HyClone, Logan, UT, USA), 1% antibiotic solution (100 U/mL penicillin and 100 μg/mL streptomycin, Symbios), 50 μM 2-mercaptoethanol (Sigma-Aldrich), and 20 ng/mL rmGM-CSF], and then placed into wells of a six-well plate and incubated at 37 °C in the presence of 5% CO_2_. The growth medium was added sequentially on day 3 of culture and replaced partially on day 6 of culture. On day 8 of culture, BMDCs were enriched using MACS CD11c^+^-labeled magnetic beads (Miltenyi Biotec, Auburn, CA, USA). Cell viability was above 95%, as determined by the 0.4% trypan blue exclusion test. 

### 4.4. Infection of BMDC Cultures

After MACS separation, CD11c^+^ cells were placed at 2 × 10^5^ or 5 × 10^5^ on a coverslip in a 24-well culture plate and infected with ECTV at M.O.I. = 1. After 1 h of virus adsorption (37 °C, 5% CO_2_), cells were cultured in the absence or presence of 1 µg/mL LPS (*Escherichia coli* B5:0111; Sigma-Aldrich) as a positive control of cell maturation. The negative control for the experiment was uninfected/unstimulated cells suspended in R-10 medium without GM-CSF. At 4, 8, 12, 18, and/or 24 h post-infection (hpi), the cells were fixed in 2% PFA. 

### 4.5. Immunofluorescence Staining

To characterize adhesion structures in BMDCs (CD11c^+^) under different experimental conditions, F-actin was stained with phalloidin conjugated either with FITC or Rhodamine Red-X (both from Sigma-Aldrich). Vinculin and paxillin were stained using unlabeled primary antibodies obtained from Sigma-Aldrich. Primary antibodies were detected with secondary anti-mouse or anti-rabbit IgG antibodies conjugated to FITC or Rhodamine Red-X (Jackson ImmunoResearch Laboratories, West Grove, PA, USA). Hoechst 33342 dye (Sigma-Aldrich) was used to label cellular and/or viral dsDNA. ECTV antigens, α-tubulin, acetylated α-tubulin, and mitochondria were stained, as previously described [46,47]. Microscopic slides were embedded in ProLong Gold Antifade Mountant (Thermo Fisher Scientific, Waltham, MA, USA). 

### 4.6. Fluorescence Microscopy Analysis

Fluorescence microscopy analysis was performed on at least 50 or 100 randomly selected single cells per slide, per experimental condition and per the experiment. Depending on the experimental condition, 100 randomly selected single cells were captured on 30–100 images. In the case of mock-infected cells, usually 30 images were analyzed per condition (per experiment), with 3–4 cells per image. In the case of ECTV-infected and/or LPS-treated cells, usually 50–100 images were analyzed per condition (per experiment) with a maximum of 2 cells per image. The reason to choose (by random) single cells was to avoid cells touching each other, which would negatively affect the determination of the number of FAs per cell. The length and width of FAs were counted in at least 250 FAs in random cells (all FAs per single random cell) per condition per experiment from three independent experiments. All microscopic analyses in ECTV-treated samples were performed on cells with clear signs of infection (presence of viral factories and/or progeny viral particles). Cells and adhesive structures were examined using an Olympus BX60 fluorescence microscope (Olympus) equipped with a PROMICAM 3-5CP camera and QuickPHOTO 2.3 software (Promicra, Prague, Czech Republic). Image analysis was performed using Cell^F 1.2 software (Olympus, Tokyo, Japan) and ImageJ 1.52a software (NIH, Bethesda, MD, USA). Brightness and contrast were adjusted using CellSens Dimension 1.9 software (Olympus) or Adobe Photoshop CS2 (Adobe Systems Inc., San Jose, CA, USA). Length and width, or FAs, were measured using ImageJ.

### 4.7. Wound Healing Assay

The motility of BMDCs was assessed using a wound-healing assay. Cells were placed on a coverslip in a 24-well culture plate to obtain a monolayer (5 × 10^5^ cells/well). Cells were then infected with ECTV for 1 h. After virus adsorption to the cells, a scratch running through the center of the slide was made on the cell monolayer using a sterile micropipette tip. Subsequently, the medium was replaced to wash off the floating cells, and LPS was added to the selected combinations (as described above). Cells were incubated for 24 h at 37 °C in a humidified atmosphere of 5% CO_2_ in air.

Wound overgrowth by BMDCs was observed at 0 and 24 h of culture using an Olympus IX71 inverted microscope (Olympus) equipped with a PROMICAM 3-5CP camera and QuickPHOTO software (Promicra). Cell migration was quantified as the difference between the number of cells within the wound at 24 h and 0 h. 

### 4.8. Transwell Migration Assay 

The transwell migration assay was used to study the migratory response of BMDCs to a culture medium derived from LPS-stimulated BMDCs. Cells were uninfected or infected with ECTV, unstimulated or stimulated with LPS for 6 h, and then subjected to the Transwell migration assay. An amount of 1 × 10^5^ cells were transferred in the upper chamber of a Transwell apparatus (5 µm pore size, Corning Inc., Lowell, MA, USA), while medium derived from LPS-stimulated BMDCs was added in the lower chamber. After 12 h of incubation at 37 °C, migrated cells were stained on the membrane or the bottom of the lower chamber with crystal violet, photographed, and counted under the inverted IX70 microscope (Olympus, Tokyo, Japan). 

### 4.9. Multicolor Immunophenotyping and Flow Cytometry

For immunophenotypic characterization, uninfected or ECTV-infected BMDCs were left unstimulated or stimulated with LPS for 24 h in a 24-well plate at a density of 1 × 10^6^ cells/well. After harvesting with 3 mM EDTA in PBS, BMDCs were blocked with 30% FBS in PBS and stained with the following monoclonal antibodies (mAb): anti-MHC II-PerCP-Cy5.5 (BioLegend, San Diego, CA, USA), anti-CD80-APC, and anti-CD86-BV711 (both from BD Biosciences, San Jose, CA, USA). After 30 min incubation and two washing steps, the cells were subjected to flow cytometry analysis using BD LSR Fortessa apparatus with BD FACSDiva 7.0 software (Becton Dickinson and Company, San Jose, CA, USA). An amount of 2 × 10^4^ events were collected for each experimental sample, and data were presented as mean fluorescent intensity (MFI). 

### 4.10. Scanning Electron Microscopy (SEM)

BMDCs grown on microscopic slides were fixed for 60 min with 2.5% glutaraldehyde in phosphate buffer, post-fixed for 60 min with 1% osmium tetroxide in phosphate buffer, and dehydrated in the ethanol and acetone series. The specimens were then dried using a CPD 7501 critical point drier (Polaron, Hatfield, PA, USA), coated with a gold layer in a JFC-1300 sputter-coater (JEOL, Tokyo, Japan), and analyzed using an FEI Quanta 200 environmental scanning electron microscope (ESEM) with the EDAX EDS system (FEI, Tokyo, Japan).

### 4.11. Statistical Analysis

Results are presented as arithmetic means ± standard deviation (SD) from at least three independent biological replicates. Statistical analysis was performed using Statistica 13.2 (StatSoft Inc., Tulsa, OK, USA) using paired or unpaired Student’s *t*-test. The correlation was performed using the Pearson coefficient. Statistical significance was assessed at * *p* < 0.05; ** *p* < 0.01; *** *p* < 0.001.

## Figures and Tables

**Figure 1 ijms-25-00558-f001:**
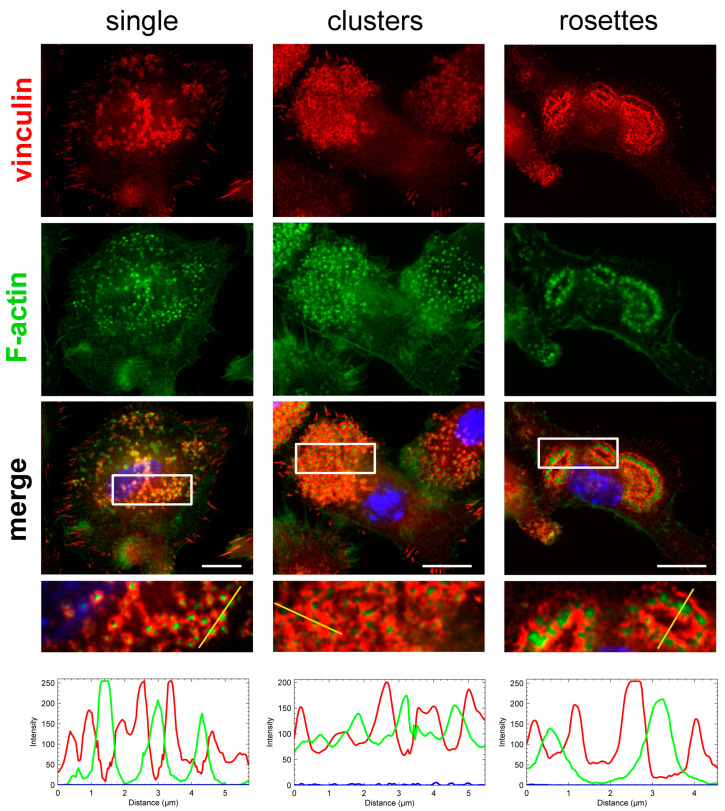
Types of podosomes in BMDCs. The fixed cells were permeabilized and stained for vinculin (red fluorescence), F-actin (green fluorescence), and nuclear DNA (Hoechst 33342, blue fluorescence). The magnified images are of the boxed regions. The fluorescence intensity of vinculin (red line), F-actin (green line), and DNA (blue line) was measured along the yellow line. Scale bars: 10 µm.

**Figure 2 ijms-25-00558-f002:**
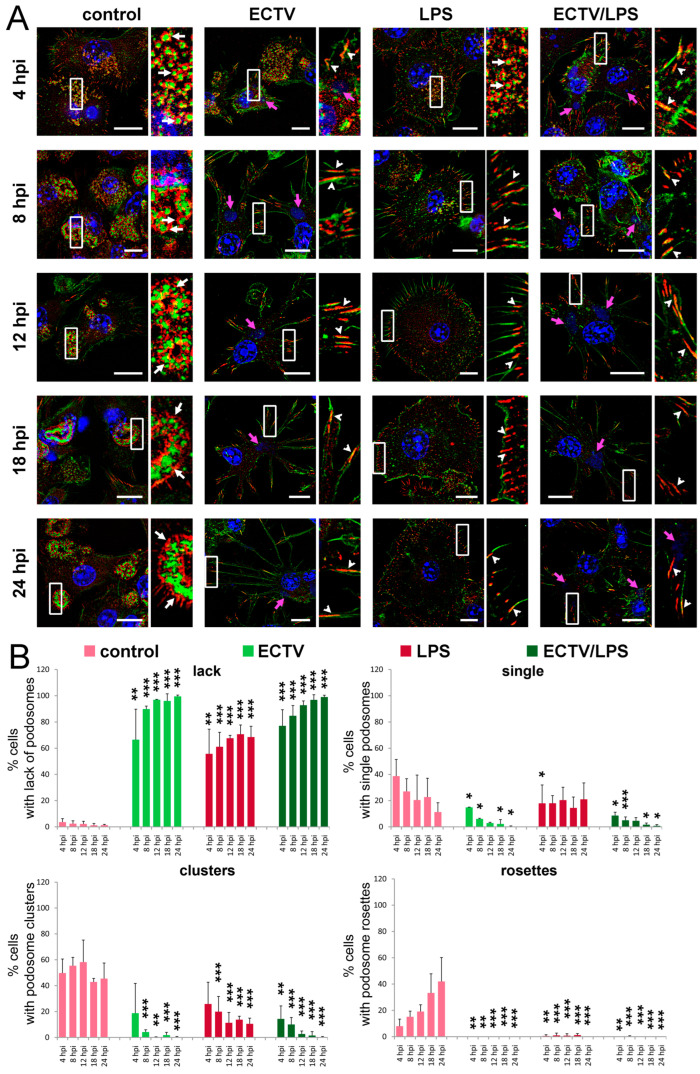
ECTV infection leads to podosome dissolution in BMDCs. Fluorescence microscopy analysis of podosomes and focal adhesions (FAs) in DCs during ECTV replication cycle. (**A**) Cells uninfected, ECTV-infected, or/and lipopolysaccharide (LPS)-treated were stained for actin (green fluorescence), vinculin (red fluorescence), and DNA (blue fluorescence) at 4, 8, 12, 18, and 24 h post-infection (hpi). The magnified images are of the boxed regions. Arrows—podosomes (white), viral factories (pink; arrowheads—FAs. Scale bars: 10 µm. (**B**) The mean percentage of cells with no podosomes or having single, clusters, or rosettes podosomes in uninfected or ECTV-infected cells, untreated or treated with LPS. The percentage of cells displaying no podosomes or different types of podosomes was counted in at least 100 random cells per condition per experiment, and an average (with standard deviation (SD)) of three experiments is shown; * *p* < 0.5, ** *p* < 0.01, *** *p* < 0.001.

**Figure 3 ijms-25-00558-f003:**
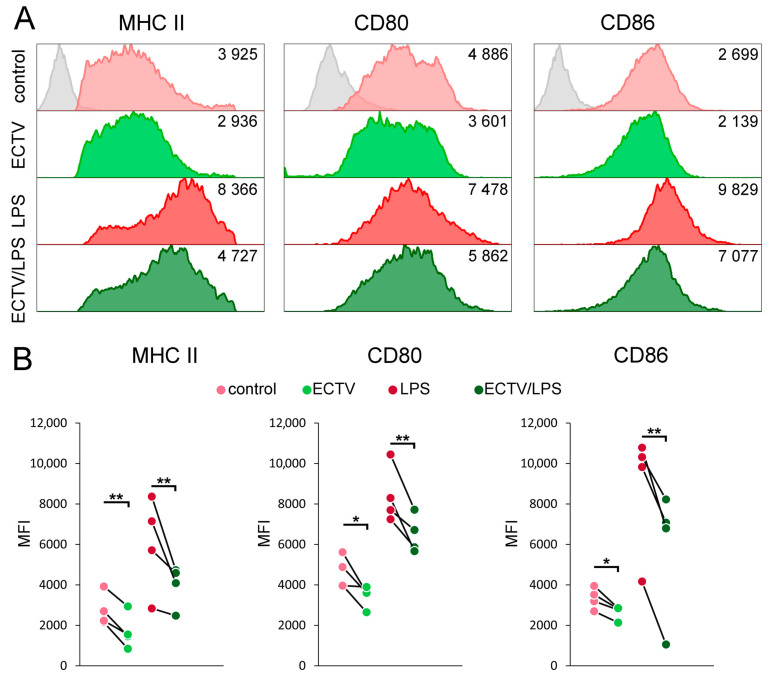
ECTV infection inhibits the maturation of BMDCs. (**A**) Representative histograms showing the major histocompatibility complex class II (MHC II), CD80, and CD86 expression on control or ECTV-infected BMDC, untreated or treated with lipopolysaccharide (LPS) at 24 hpi. Numbers represent the MFI value for a given marker. Grey histograms—isotype controls. (**B**) Graphs show individual data of mean fluorescence intensity (MFI) of MHC II, CD80, and CD86 molecules from four independent experiments. Significant differences were indicated by horizontal bars between two sets of data (* *p* < 0.05, ** *p* < 0.01).

**Figure 4 ijms-25-00558-f004:**
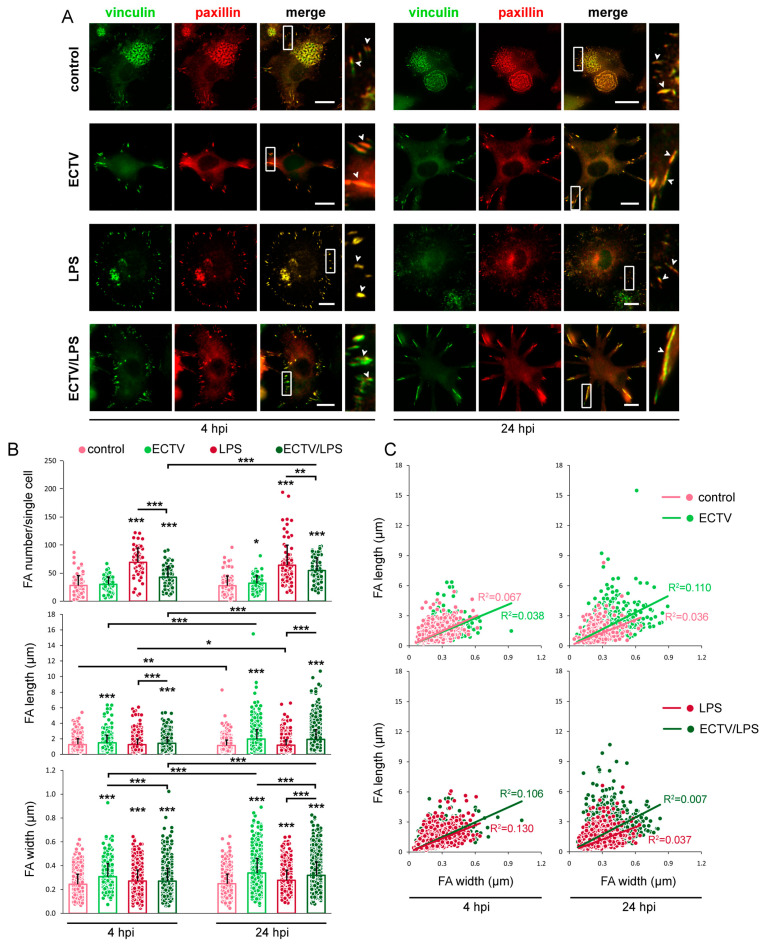
ECTV infection induces elongation and widening of focal adhesions (FAs) in BMDCs. (**A**) Fluorescence microscopy analysis of FAs in uninfected or ECTV-infected cells, untreated or treated with lipopolysaccharide (LPS). The cells were stained for vinculin (green fluorescence) and paxillin (red fluorescence) at 4 and 24 h post-infection (hpi). The magnified images are of the boxed regions. Arrowheads—FAs. Scale bars: 10 µm. (**B**) The number of FAs/cell and the mean length and width (both in µm) of FAs in uninfected or ECTV-infected cells, untreated or treated with LPS at 4 and 24 hpi. The number of FAs/cell was counted in at least 50 random single cells per condition per experiment from three independent experiments. The length and width of FAs were counted in at least 250 FAs in random cells (all FAs per single random cell) per condition per experiment from three independent experiments. Each point represents individual data, with the bar indicating the mean values of three independent experiments; * *p* < 0.5, ** *p* < 0.01, *** *p* < 0.001. (**C**). The correlation between length and width of FAs. Each point represents the length vs. width of a single FA of at least 200 FAs per condition per experiment from three independent experiments.

**Figure 5 ijms-25-00558-f005:**
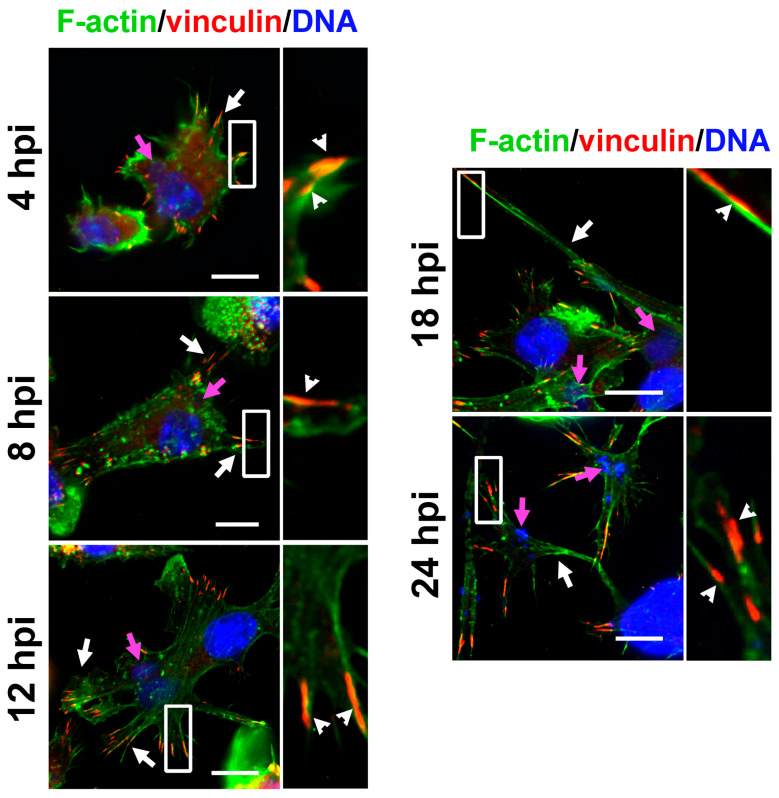
ECTV infection promotes the formation of focal adhesions (FAs) in long cellular extensions of BMDCs. Fluorescence microscopy analysis of dynamics of formation of long cellular extensions in ECTV-infected cells. The cells were stained for F-actin (green fluorescence), vinculin (red fluorescence), and DNA (blue fluorescence) at 4, 8, 12, 18, and 24 h post-infection (hpi). The magnified images are of the boxed regions. Arrows—cellular extensions (white), viral factories (pink); arrowheads—FAs. Scale bars: 10 µm.

**Figure 6 ijms-25-00558-f006:**
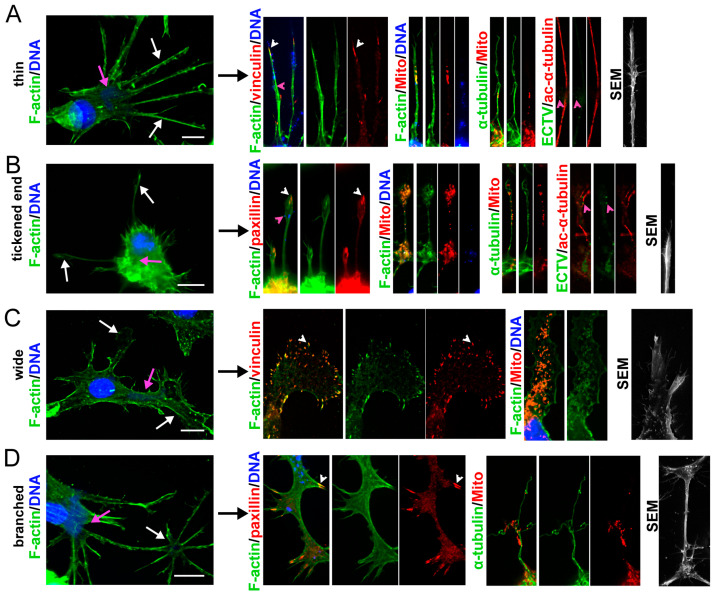
Types of long cellular extensions formed by ECTV-infected cells and localization of FAs within them. (**A**) Thin, (**B**) tickened end, (**C**) wide, and (**D**) branched long extensions. The cells were stained for F-actin, α-tubulin or ECTV antigens (green fluorescence), vinculin, paxillin, mitochondria (Mito) and acetylated-α-tubulin (ac-α-tubulin) (red fluorescence), and DNA (blue fluorescence), or analyzed using scanning electron microscopy (SEM) at 24 h post-infection (hpi). The magnified images are of the boxed regions. Arrows—long cellular extensions (white), viral factories (pink); arrowheads—FAs (white), viral particles (pink). Scale bars: 10 µm.

**Figure 7 ijms-25-00558-f007:**
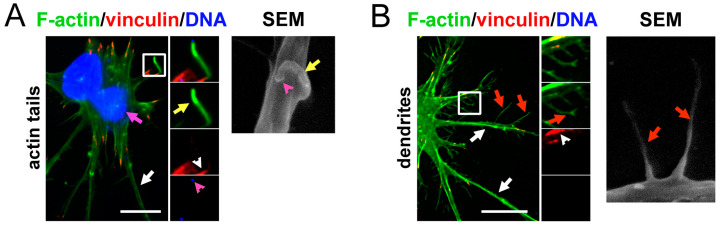
Types of short cellular extensions formed by ECTV-infected cells and localization of FAs within them. (**A**) Actin tails and (**B**) dendrites. The cells were stained for F-actin, α-tubulin or ECTV antigens (green fluorescence), vinculin, paxillin, mitochondria (Mito) and acetylated-α-tubulin (ac-α-tubulin) (red fluorescence), and DNA (blue fluorescence), or analyzed using scanning electron microscopy (SEM) at 24 h post-infection (hpi). The magnified images are of the boxed regions. Arrows—long cellular extensions (white), actin tails (yellow), dendrites (red), viral factories (pink); arrowheads—FAs (white), viral particles (pink). Scale bars: 10 µm.

**Figure 8 ijms-25-00558-f008:**
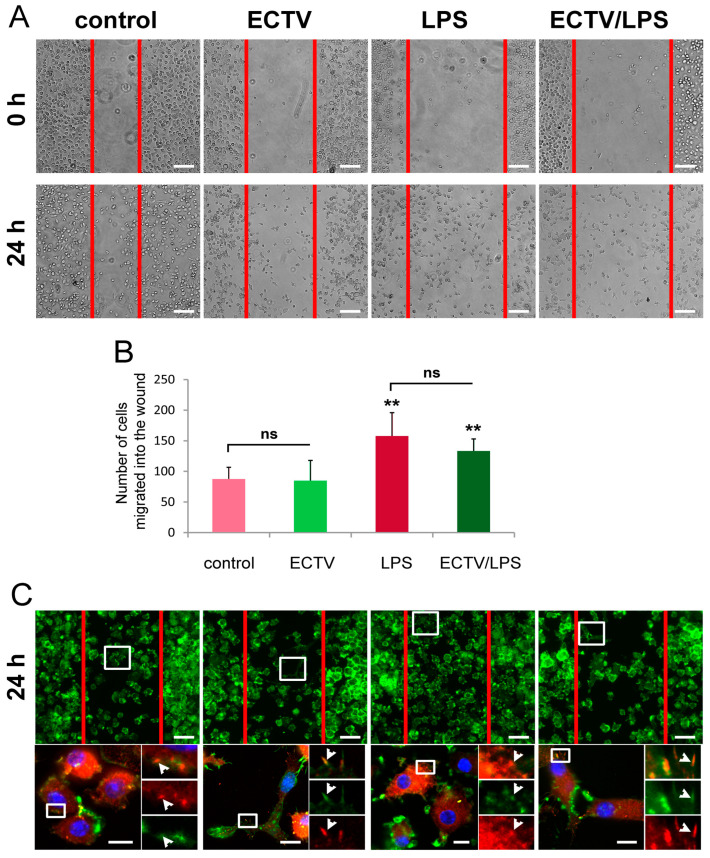
ECTV infection does not influence the migratory rate of ECTV-infected BMDCs in a wound healing assay. (**A**). Representative images of BMDC migration into the scrach region (outlined in red) at 0 and 24 h. (**B**) Graphs show the mean number of cells migrated into the wound with standard deviation (SD) from three independent experiments; ** *p* < 0.01, n.s.—non-significant. Scale bars: 100 µm. (**C**) Fluorescence microscopy analysis of the presence of focal adhesions (FAs) in BMDCs migrated into the wound area (outlined in red). The cells were stained for F-actin (green fluorescence), vinculin (red fluorescence), and DNA (blue fluorescence) at 24 h post-infection (hpi). The magnified images are of the boxed regions. Arrowheads—FAs. Scale bars: 50 µm (**upper** panel) or 10 µm (**lower** panel).

**Figure 9 ijms-25-00558-f009:**
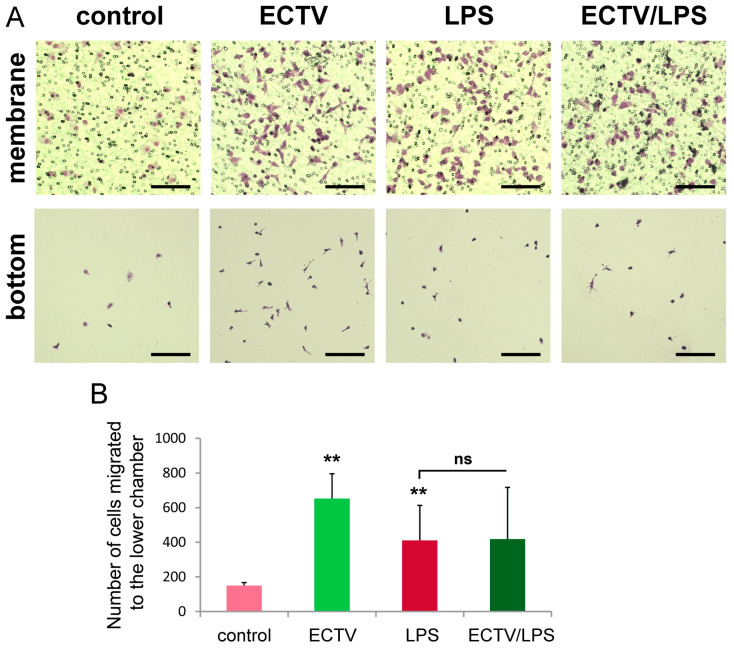
ECTV infection accelerates the motility of infected BMDCs toward supernatants derived from the LPS-conditioned BMDCs. (**A**). Representative microscopic images of cells that migrated through the transwell and remained on the underside of the membrane or migrated to the bottom of the lower well in the migration assay (crystal violet stain). Scale bars: 50 µm (upper panel) or 100 µm (lower panel). BMDCs were uninfected or infected with ECTV for 6 h and then subjected to the Transwell migration assay. After incubation for 12 h, migrated cells were stained on the membrane or on the bottom of the lower chamber with crystal violet and photographed. (**B**) Graphs show the mean number of cells migrated to the lower chamber in a Transwell migration assay with standard deviation (SD) from three independent experiments; ** *p* < 0.01, n.s.—non-significant.

## Data Availability

All data generated or analyzed during this study are included in this published article.

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
