# Peer review of "Ectromelia Virus Affects the Formation and Spatial Organization of Adhesive Structures in Murine Dendritic Cells In Vitro"

_ijms, 2023, doi:10.3390/ijms25010558_

Round 1

Reviewer 1 Report

Comments and Suggestions for Authors

In this manuscript, the authors investigated how Ectromelia virus infection modifies the migration and the maturation of natural-host bone marrow-derived dendric cells, focusing particular attention on the formation of podosomes and focal adhesions.

The article is well-written, and the rationale of all experiments is well-explained and clear.

In my opinion, there are only minor concerns:

Please, add the unit on y-ass of the graphs in Figures 2B, 6B, and 7B.

In lines 440-450, the authors described that different pathogens may induce DC maturation, but have different effects on podosome formation. In the case of ECTV, the infection does not induce DC maturation, but it is associated with podosome dissolution. Are there other pathogens, described in the literature, that can do the same effect? 

Reviewer 2 Report

Comments and Suggestions for Authors

Dear Editor,

In the manuscript “Ectromelia Virus Affects the Formation and Spatial Organization of Adhesive Structures in Murine Dendritic Cells in VitroBiernacka S. and co-authors demonstrate and discuss the effect of ECTV infection on the formation of podosomes and focal adhesions by mouse Dendritic cells (BMDC). The topic by itself is interesting but the experimental material and its presentation, I think,  require improvement.  The manuscript discusses the effect of ECTV infection of  dendritic cells on the structure and formation of podosomes and Focal Adhesions (FA) however I did not find any quantification of the efficiency of infection and any applicable measurements of the virus load and replication in those cells. It is not shown at all if those cells that are discussed in the analysis as ECTV infected indeed have viruses and if it is a productive infection. Also, I am very confused by the logic of the manuscript where the main experimental part is observing the formation of podosomes during culture of dendritic cells  on the stress surface conditions on the glass but then conducting wound healing test or migration test in TC-treated plates, where authors do not detect those structures (line 343).  Why would not authors conduct the same tests to observe the formation of podosomes and FAs using  more physiologically relevant ECM-treated surfaces? Can it be considered as the artificial structure and as  result of artificial conditions of in vitro culture which cell biology is very well trying to avoid by using a lot of different natural and synthetic ECM coating? How all those observations can be relevant for the biological interactions between viruses and dendritic cells in vivo and in tissues?

There is also a list of serious comments below:

1.      The discussion of the structure and shapes of podosomes in part 2.2 of results, and Figure 1, and Figure 2 (also for Figure 5) show no information on how those images were analyzed to measure all values for the podosomes and FA structures. What were the criteria to classify structures as “rosettes”, or  “clusters” , or others for Figure 2B.  The graph Figure 2B is very hard to read to understand the % of each type of structure. It would be better to add a table with numbers and SD for each barograph.

2.      Figure 3.A. needs both X-axes and Y-axes with labels and scale. Figure 3B is labeled as showing mean ± SD but it does not show SD or error bars were not placed to the dots.

3.      For discussion of statistical significance – it is not clear how cells were “randomly selected” for all those measurements (Figures 1,2,4 and 5).  What was the total number of images of cells in cases where  100 cells per condition (line 93, Figure 2) or 50 cells (line 267 figure 4) were selected? Why the numbers 100 or 50 were considered sufficient for statistical representation?  Also for the same Figure 4 also “random”  250 and later  200 of FA per conditions  (lines 269 and 272) were selected. I think the results are missing a description how those selections of images for statistics were  done.

4.      Figure 4B also would benefit from showing the median and SD numbers under the plots to make it better interpretable because each of the 3 graphs has a different scale.

5.      Figure 4C can be just mentioned but not shown – because it does not give any information except that there were no correlations at any condition tested,  or if there is some useful information it  needs to be mentioned in line 255-256.

6.      There is no information on how the measurements in Figure 4 are related to the changes depicted in Figure 5 branches, thin, thick, and wide. Also, it is important to show the measurements that were used to classify these structures (same issues as for the identification and classification of podosomes.

7.      Figure 5 B some small images and 5D images - are too small and hard to see. Also, the way they are presented outside of context does not help to see the significance of those illustrations.

8.      The section line 305- line 322 in results -in reality, is a discussion with speculations and  no real data. Maybe it can be better rearranged in the Discussion section of the manuscript.

Round 2

Reviewer 2 Report

Comments and Suggestions for Authors

Dear Authors,

Thank you, for your detailed responses and very careful corrections. The one minor issue that remains uncorrected  - is the missing  “scale bars” for each image with cells i.e. Figure 1, 2A, 4A, 5,6 & 7 to correct for zooming of fragments of images the same way as it was used for the cell images in publication (Szulc-Dąbrowska, L.; Struzik, J.; Ostrowska, A.; Guzera, M.; Toka, F.N.; Bossowska-Nowicka, M.; Gieryńska, M.M.; Winnicka, A.; Nowak, Z.; Niemiałtowski, M.G. Functional Paralysis of GM[1]CSF-Derived Bone Marrow Cells Productively Infected with Ectromelia Virus. PLoS One2017, 12, doi:10.1371/JOURNAL.PONE.0179166).

Thank you also for providing very detailed references from your manuscript in response to my comments about the classification of the structures of podosomes  (Figure 1) and morphology of cellular pseudopods ( Figure 6 in the revised version) in the cellular images. I  was thinking more about using image analysis software like  Matlab or ImageJ that adds unbiased analytical values to your keen observations and detailed descriptions. I am using those methods intensely in my work and think those applications can be very useful in your research on cellular morphology using fluorescent imaging approaches.
